# Targeting the Tumor Microenvironment in Breast Cancer: Prognostic and Predictive Significance and Therapeutic Opportunities

**DOI:** 10.3390/ijms242316771

**Published:** 2023-11-26

**Authors:** María A. Domínguez-Cejudo, Ana Gil-Torralvo, Mónica Cejuela, Sonia Molina-Pinelo, Javier Salvador Bofill

**Affiliations:** 1Institute of Biomedicine of Seville (IBiS), HUVR, CSIC, Universidad de Sevilla, 41013 Seville, Spainsmolina-ibis@us.es (S.M.-P.); 2Andalusian—Roche Network Mixed Alliance in Precision Medical Oncology, 41092 Sevilla, Spain; 3Medical Oncology Department, Virgen del Rocio Hospital, 41013 Seville, Spain

**Keywords:** breast cancer, tumor microenvironment, biomarkers, targeted therapy

## Abstract

Breast cancer is one of the most prevalent tumors among women. Its prognosis and treatment outcomes depend on factors related to tumor cell biology. However, recent studies have revealed the critical role of the tumor microenvironment (TME) in the development, progression, and treatment response of breast cancer. In this review, we explore the different components of the TME and their relevance as prognostic and predictive biomarkers in breast cancer. In addition, techniques for assessing the tumor microenvironment, such as immunohistochemistry or gene expression profiling, and their clinical utility in therapeutic decision-making are examined. Finally, therapeutic strategies targeting the TME are reviewed, highlighting their potential clinical benefits. Overall, this review emphasizes the importance of the TME in breast cancer and its potential as a clinical tool for better patient stratification and the design of personalized therapies.

## 1. Introduction

Breast cancer (BC) is a prevalent and lethal disease, affecting more women than men and exhibiting geographical disparities [1]. It is considered a heterogeneous disease, and its classic prognostic assessment has relied on clinical and histopathological criteria, including lymph node involvement, tumor size, and tumor cell differentiation [2].

Immunohistochemical (IHC) markers such as estrogen receptor (ER), progesterone receptor (PR), and human epidermal growth factor receptor 2 (HER2) have refined subtype differentiation and influenced treatment decisions. Proliferation markers like Ki67 also offer prognostic value [3,4,5]. Advances in molecular biology have allowed platforms like Oncotype to provide prognostic and predictive information for the selection of patients who should undergo adjuvant chemotherapy [6]. Furthermore, in recent years, the targeted sequencing of specific genes, such as phosphatidylinositol-4,5-bisphosphate 3-kinase catalytic subunit alpha (PIK3CA) or estrogen receptor alpha 1 (ESR1), has been introduced in clinical decision-making as their mutational profiles provide predictive information regarding the response to certain treatments.

While genetic and epigenetic factors significantly contribute to breast carcinoma, the tumor microenvironment (TME) plays a pivotal role in cancer development, progression, and metastasis. The TME comprises diverse cell types (stromal, immune, endothelial, and adipocytes) that interact with cancer cells, fostering tumor growth and functional support. [7]. From a historical perspective, the first evidence of the decisive role of the stroma in tumor development came from studies on tumor neovascularization, in which new blood vessels are formed in response to pro-angiogenic signals secreted from tumor cells. Since then, numerous studies have contributed to the characterization of the TME, further complicating the already challenging task of understanding and treating cancer. The aim of our review is to describe the components of the TME and their clinical role as prognostic and predictive factors in breast cancer, as well as therapeutic options that seek to maximize TME–cancer cell communication.

## 2. Components of the Breast Cancer Microenvironment and Their Value as Prognostic and Predictive Factors

The TME is composed of a complex network of different cell types. It consists of infiltrating immune cells such as lymphocytes (T cells and B cells), mast cells, and antigen-presenting cells (macrophages and dendritic cells), as well as tumor-associated fibroblasts, endothelial cells, the extracellular matrix (ECM), and other stromal components (Figure 1). Current data support the clinical benefit of using immune and stromal gene signatures as prognostic and predictive markers in breast cancer. In this section, we will detail the key components of the TME and its prognostic and/or predictive value across different BC subtypes. Table 1 compiles the components whose prognostic or predictive value has been tested in patient cohorts with clinical utility.

### 2.1. The Immune System in the Breast Tumor Microenvironment

The notion that the immune system not only protects the host from tumorigenesis but also shapes tumor immunogenicity is the basis for the cancer immunoediting hypothesis. This process encompasses three stages: elimination, equilibrium, and escape [8]. During the elimination stage, the immune system recognizes and eliminates the most immunogenic cancer cells through cytotoxic mechanisms. In the equilibrium stage, the tumor is held in check via immunosurveillance, although certain tumor cells manage to evade immune responses and promote tumor growth. Through this escape stage, the most aggressive clones adopt different strategies to evade immune recognition and promote the formation of an immunosuppressive TME, contributing to the development of malignant neoplasia [8]. The balance between immune components that provide protective antitumor immunity by targeting immunogenic tumor variants and those that facilitate tumor progression, shaping tumor immunogenicity, determines the magnitude of the immune response generated. This immunogenic capacity not only is critical for disease evolution and prognosis but also holds significant predictive value, influencing clinical responses to therapeutic regimens [9].

#### 2.1.1. Tumor-Infiltrating Lymphocytes (TILs)

Tumor-infiltrating lymphocytes (TILs) are white blood cells that have migrated from the bloodstream into the tumor. They comprise various immunophenotypic and functionally distinct subpopulations, including CD8+ cytotoxic T cells, CD4+ helper T cells, B cells, and natural killer (NK) cells [10]. There is evidence that the infiltrating lymphocyte population differs throughout the course of the disease, being more abundant in early stages and at the onset of metastatic disease compared to advanced multi-treated stages [11]. Furthermore, in TN and HER2+ (i.e., ER-) tumors, there is higher infiltration of TILs compared to ER+ tumors, highlighting the heterogeneity among different subtypes [12,13].

Different studies have evaluated the prognostic value of TILs using immune and hematoxylin and eosin (H&E) staining, flow cytometry, and/or gene expression analysis. The data consistently demonstrate that TIL infiltration is associated with improved clinical outcomes [10]. Tumor infiltration via TILs correlates with longer overall survival (OS) independently of other clinical–pathological parameters [14]. The improved prognostic impact of TILs is closely linked to an increase in the density of cytotoxic T lymphocytes, or CD8+ lymphocytes, mainly in ER- tumors [15]. In fact, the clinical utility of using TILs as prognostic biomarkers in TN is well established, and the International Immuno-Oncology Biomarker Working Group has provided a standardized and reproducible method for assessing TIL density in BC [16]. Although a high percentage of TILs suggests a favorable prognosis for ER- tumors, their impact on ER+ tumors remains unknown. Several studies have retrospectively studied the role of TIL infiltration in the prognosis of these patients, but to date, no positive impact has been shown on OS, disease-free survival (DFS), or breast-cancer-specific survival. In large cohorts of patients with ER+ tumors, actually, high levels of CD8+ TILs have been associated with an unfavorable prognosis [17,18].

The clinical utility of TIL infiltration is not limited to its prognostic value. Different studies have shown that assessing the quantity/density of TILs can be used as a predictive factor for specific therapeutic treatments, especially in ER- tumors. For instance, in the context of neoadjuvant chemotherapy (NAC), there is a priority to identify patients who have the potential to achieve a pathological complete response (pCR), as it is associated with excellent long-term survival [19]. In one of the earliest studies evaluating the predictive role of TILs, the authors demonstrated that the number of CD3+ TILs in pre-treatment biopsies was significantly higher in patients who achieved pCR [20]. Since then, different studies have shown that a higher number of TILs in pre-NAC biopsies correlates with higher rates of pCR [21,22,23].

In summary, TILs exhibit functional and phenotypic diversity. Therefore, a panel of parameters, including TIL counts, TIL subsets, their tumor reactivity, and functional states, should be considered to define truly “hot” tumors that may help predict a better prognosis following cancer therapy.

#### 2.1.2. Tumor-Associated Macrophages

Tumor-associated macrophages (TAMs) play a role in tumor biology by mediating tumor growth and progression, as well as contributing to therapy resistance [24]. In breast cancer, resident macrophages and the recruitment of circulating monocytes support TAM accumulation [25]. TAMs have historically been divided into two categories: M1 and M2. M1 refers to macrophages that undergo classical activation via interferon-γ (IFNγ) with either lipopolysaccharide (LPS) or TNF, whereas M2 refers to macrophages that undergo alternative activation via IL-4 [26]. M1 or M2 polarized macrophages have opposing effects on tumor progression. Evidence suggests that an increase in M1 macrophages in the TME is associated with reduced tumor aggressiveness, while an increase in M2 macrophages is related to tumor growth and poor cancer prognosis [27]. Once TAMs acquire an M2 phenotype upon interaction with cancer cells, T cells, or other cell types within the TME, the tumor progresses through adaptive immunity suppression, tissue remodeling, and angiogenesis [28,29].

In the early 1970s, Wood and Gollahon observed the presence of macrophages in the breast TME, which determined the risk of disease progression and therapeutic resistance [30]. Since then, multiple clinical studies have supported the value of enumerating TAMs for prognosis and/or predicting outcomes [31,32,33]. For instance, the work by Mahmoud, et al. confirmed the prognostic value of TAMs using a large cohort (1322) of breast cancer patients. In their survival analysis, a higher number of CD68+ macrophages predicted worse breast-cancer-specific survival and shorter DFS [33]. Several independent studies have also revealed that the intensity of macrophage infiltration is associated with ER negativity and a high mitotic rate [34,35]. This is likely due to the distinct biology of ER+/− tumors, with a more hypoxic and pro-inflammatory phenotype in ER- cancers.

The TAM status also predicts sensitivity to chemotherapy and radiation therapy. When leukocyte complexity in breast cancer tissue was evaluated, Ruffell, et al. found that, in patients who had not received chemotherapy, macrophages were predominantly present in non-adjacent normal tissues but not in breast tissues [36]. In contrast, tumors from patients undergoing neoadjuvant chemotherapy contained elevated levels of infiltrating macrophages [36,37]. Macrophage infiltration (CD68+) is associated with lymph node metastasis, adverse pathological stage, and, consequently, a higher likelihood of receiving adjuvant chemotherapy [38]. These findings align with current experimental discoveries that TAMs directly mitigate the chemotherapeutic response in BC by releasing chemoresistance factors or indirectly regulating drug resistance in cancer stem cells [39,40,41].

### 2.2. Cancer-Associated Fibroblasts (CAFs) and the Extracellular Matrix (ECM)

Cancer-associated fibroblasts (CAFs) are one of the most abundant stromal components in the TME. Multiple studies have demonstrated that CAFs play a prominent role in cancer pathogenesis, which has significant clinical implications [42]. Activated CAFs show migratory and proliferative properties, unlike their inactive counterparts. Their most distinctive feature is the high capacity of synthesis and remodeling of the extracellular matrix (ECM) during the desmoplastic reaction [43,44]. In the desmoplastic reaction, activated fibroblasts abundantly synthesize various types of collagens, hyaluronan, fibronectins, and laminins that constitute the ECM and basement membrane [45]. The deposition of these components in the tumor stroma can act as a physical barrier against immune infiltration or as a structural scaffold for intercellular interaction, thus modulating tumorigenesis [46]. Simultaneously, myofibroblasts enhance ECM turnover by producing matrix metalloproteinases (MMPs) that degrade it. ECM degradation allows the vascular endothelial growth factor A (VEGFA), which is sequestered in the ECM, to interact with VEGF receptors (VEGFRs) and, thus, promote angiogenesis [47]. CAFs also secrete large quantities of growth factors, pro-inflammatory cytokines, and chemokines to recruit immune cells, especially immunosuppressive ones, to support immune evasion [48,49]. Overall, CAFs play a pivotal role in shaping a microenvironment that favors tumor initiation, angiogenesis, dissemination, and metastasis through the production of multiple ECM proteins and regulatory molecules.

Similar to other cancer types, most studies have applied IHC to identify potential biomarkers derived from CAFs in BC. These studies suggest that the nature and quantity of CAFs have direct prognostic relevance in these patients. For instance, a higher proportion of αSMA-positive myofibroblasts has been associated with increased tumor cell proliferation and shorter recurrence-free survival and breast-cancer-specific survival [50]. Finak, et al. used laser capture microdissection to isolate fibroblasts from normal breast tissue and breast cancer to establish a “CAF gene signature” consisting of 26 genes associated with an adverse outcome in four published datasets comprising a total of 1021 cancerous tissues [51]. Researchers have also focused on specific molecular subtypes of BC. Roman-Perez, et al. defined an “active stroma signature” composed of genes involved in cell motility and fibrosis activation, which was associated with poor prognosis in 43 ER+ samples [52]. On the other hand, the study by Beck, et al. suggested that an activated stroma does not always indicate a more aggressive clinical behavior. They constructed a gene signature derived from desmoid-type fibromatosis, a type of soft tissue tumor. The signature was applied to identify a fibrotic stromal reaction associated with a favorable prognosis in ER+ breast cancer patients [53]. Other CAF markers belong to common growth factor pathways, such as TGFβ and hedgehog (Hh) signaling pathways, known for paracrine signaling between epithelial cells and the microenvironment. In a cohort of 252 breast cancer patients from a randomized trial, the expression of the TGFβ type 2 receptor (TGFBR2) via CAFs was an independent marker of prolonged survival [54]. Gli1 (GLI Family Zinc Finger 1) is a protein belonging to the Hh signaling pathway. Cytoplasmic expression of Gli1 in CAFs was associated with shorter survival in a cohort of 279 invasive breast cancer patients. This association was even more pronounced when the Hh ligand was concurrently expressed via cancer cells, suggesting a clinically relevant paracrine mechanism [55].

The evaluation of CAFs in BC can also serve as a predictive factor. In this regard, Yamashita, et al. identified that α-SMA expression in myofibroblasts is not only associated with a worse prognosis but also serves as an independent predictor of metastasis in patients with invasive breast cancer [56]. Interestingly, a pilot study for the detection of circulating CAFs in the peripheral blood of patients with advanced breast cancer determined that circulating CAFs were present in 88% of patients with metastasis compared to 33% of patients with localized tumors [57].

In summary, CAFs exhibit heterogeneous subtypes with diverse origins and markers, holding significant potential in clinical diagnostics. Therefore, studying the phenotypic, molecular, and functional characteristics of CAFs that prevail in breast tumors is crucial, as it can provide valuable insights for personalized treatment approaches.

### 2.3. The Tumor Vasculature

Angiogenesis is a fundamental biological process that involves the formation of new blood vessels from pre-existing ones. It plays a crucial role in different physiological and pathological conditions, including wound healing, embryonic development, and tumor growth. In the context of cancer, angiogenesis is a critical factor that supports tumor progression, invasion, and metastasis. The process of angiogenesis involves several steps. First, endothelial cells, the building blocks of blood vessels, are activated and start to migrate towards the tumor under the influence of pro-angiogenic factors. Then, these endothelial cells proliferate and form tube-like structures, eventually connecting to the existing vasculature to create functional blood vessels that supply nutrients and oxygen to the tumor. However, as they grow and expand, tumors face a state of hypoxia or low oxygen levels.

Hypoxia induces the expression of hypoxia-inducible factor (HIF), which upregulates a series of oncogenes associated with an aggressive neoplastic cell phenotype [58]. In particular, the overexpression of HIF-1α protein has been identified in different types of tumors, where high levels influence the growth rate and metastatic potential of these tumors. In BC, the frequency of HIF-1α-positive cells increases with the clinical stage and is associated with a worse prognosis [59].

Interestingly, evidence from transgenic models indicates that the acquisition of this angiogenic phenotype occurs early in development and may have predictive implications [60]. These experimental models are supported by findings in human tissues, revealing that normal breast tissue adjacent to malignant breast tissue induces angiogenesis at a rate twice as high as non-neoplastic breast tissues, suggesting the angiogenic shift occurs prior to detectable morphological changes [61]. Using microvessel density as a surrogate for angiogenesis associates benign lesions that have high vascular density with a greater risk of developing breast cancer [62]. Moreover, quantifying angiogenesis has been proposed as a potential predictor of in situ cancer progression [63] and treatment response [58]. It has also been demonstrated to have a direct correlation with the presence of bone marrow micrometastases [64] and overall survival [65].

### 2.4. Adipocytes

Adipocytes are the primary cellular component of adipose tissue, and they play a crucial role in maintaining the energy balance. The dysregulation of adipocyte function leads to overweight and obesity. Various studies have demonstrated a relationship between obesity and a worse prognosis for breast cancer. This includes a prospective study of nearly 500,000 women, which established a progressive escalation in the risk of breast cancer mortality with each successive increase in BMI category [66]. Other studies have shown a positive correlation between obesity and breast cancer recurrence [67], shorter disease-free survival, and lower rates of overall survival [68].
ijms-24-16771-t001_Table 1Table 1Components of the tumor microenviroment.
Cell Type and SubtypeFunction and LocationPrognostic and/or Predictive ValueMarkersRefsImmune systemLymphocytes T lymphocytes and cytotoxic T cellsHelper T cells, and regulatory T cells,Memory T cellsOnce activated, naive CD4+ and CD8+ T lymphocytes, present in the blood and lymphoid organs, generate different cell subsets to perform cytotoxic functions. CD8+ subsets: Tc1 and Tc2. CD4+ subsets: Th1, Th2, Th17, and Treg (Foxp3). Following exposure to an antigen, a small subset of effector T cells differentiate into memory cells. Memory T cells maintain their antigen specificity and help to amplify the immune response during antigen re-exposure.NACAll subtypes: higher TIL infiltration increases pCR ratesAdjuvant treatment TNBC: increase the DFS for each 10% increment of sTILsTNBC and ER-/Her2+: the presence of CD8+ cells increases breast-cancer-specific survivalER+: the presence of CD8+ decreases DFST lymphocyte: CD69, CD25, CD45RO, OX40, 4-1BBL, CD95, Granzyme B, and Perforin, CD44Memory T lymphocyte: CCR7, CD62L, CD45RO, CD45RA, CD95, CD127, CD28, and Granzyme B[13,14,16,17,20,21,22,23,24,25,26]MacrophagesPhagocytic cells that consume foreign pathogens and cancer cells. Stimulate the responses of other immune cells.Migrate from blood vessels into tissues.Increased TAMs decrease the OSIncreased TAMs CD68+ decrease the DFS and breast-cancer-specific survivalCD68, CD64, CD11b, and colony-stimulating factor 1 receptor[34,35,36]Other stromal componentsFibroblastsFibroblasts create and maintain anatomically diverse extracellular-matrix (ECM)-rich connective tissues to support a broad range of essential organ functions. Fibroblasts provide essential niches and positional information for neighboring cells via biochemical cues in the ECM and the regulated secretion of soluble mediators such as cytokines, growth factors, and metabolites. In addition, fibroblasts serve as the progenitors of specialized mesenchymal cell types, such as bone-forming osteoblasts or lipid-filled adipocytes during embryonic development, adult homeostasis, and injury, repair, and remodeling.Higher αSMA+ fibroblasts decrease recurrence-free survival and breast cancer-specific survival and serve as a predictor of metastasis; the CAF gene signature is associated with an adverse outcomeThe ER+: active stroma signature decreases the OS TGFBR2+ cells increase the DFSCytoplasmic Gli1: shorter survival Fibroblast activation protein α, α smooth muscle actin, microfibril-associated protein 5, collagen type XI alpha 1 chain, tenascin-C, podoplanin, integrin α11β1, and neural/glial antigen 2[53,55,57,59]VasculatureTumor angiogenesis entails the development of new blood vessels from established vascular beds. Hypoxia and nutrient deprivation trigger an “angiogenic switch” to allow the tumor to progress. The resulting vessel network is leaky, unorganized, and ill-perfused, determining how cancer cells escape anticancer immunosurveillance, metastasize, and respond to therapy.High HIF-a1 expression, shorter DFS and OS. Increased MVD decrease DFS and OSVEGFR, integrin αvβ3, CD44-related antigen, fibronectin ED-B domain, endoglin, endosialin, E-selectin, and vascular cell adhesion molecule 1[62,68]AdipocytesAdipose tissue is a connective tissue mainly composed of adipocytes. Adipocytes are energy-storing cells that contain large globules of fat known as lipid droplets surrounded by a structural network of fibers. At the cancer invasion front, adipocytes undergo lipolysis and transform into CAAs. CAAs can secrete a variety of adipokines and release free fatty acids and exosomes to cancer cells for metabolic reprogramming.Increased BMI higher risk of mortality.Obesity decreases OS Leptin, Hoxc8, Hoxc9, Ucp1, CIDEA, PRDM16, Zic1, Lhx8, Eva1 Epsti1 Cd137, Tmem26, Tbx1, Cited1, and Shox2[66]


Obesity-induced inflammation is an essential mechanism in the development and invasion of breast cancer [69]. Cancer-associated adipocytes (CAAs) are characterized by a dysfunctional phenotype and a more aggressive secretome with a direct implication in the reprogramming of metabolism and ECM remodeling. In addition, adipocytes have been shown to influence macrophage polarization in the adipose tissue and in the TME. Fatty acids’ uptake and oxidation were associated with M2 anti-inflammatory, immunosuppressive, and tumor-promoting polarization [70].

## 3. The TME as a Therapeutic Target

As early as 1889, the pioneering cancer researcher Stephen Paget proposed the “seed and soil” theory and suggested that cancer cells (the seed) can only induce tumor formation in the presence of a favorable microenvironment (the soil) [71]. While cancer prevention and intervention strategies have mainly focused on the intrinsic factors of cancer cells, in the last decade, a burst of knowledge about how the TME interacts with tumor cells has driven research into a new paradigm of cancer treatment: targeting the tumor stroma. Numerous studies have focused more on perivascular cells, endothelial cells, fibroblasts, and different active immune cells present in the TME. In fact, one of the most significant advances in cancer treatment has been the development of immune checkpoint inhibitors (ICIs) that have demonstrated durable responses and improved survival in multiple solid malignancies [72]. This growing interest in understanding the critical role of the TME in the development and progression of breast cancer has prompted the exploration of more effective and personalized therapeutic approaches to fight this devastating disease. Table 2 compiles the approved treatments and phase III clinical trials that have harnessed the TME to combat tumors.

### 3.1. Exploiting the Immune System for Therapeutic Benefit

#### 3.1.1. Immune Checkpoint Inhibitors

Immune checkpoints are essential for regulating the immune response and preventing tissue damage. When proteins on the surfaces of T cells bind to proteins in other cells, like tumor cells, these checkpoints become activated to restrict the immune response. However, this signaling can also create an environment that allows tumor cells to evade destruction via the immune system. For over a decade, it has been shown that blocking these immune checkpoints with monoclonal antibodies can trigger effective anti-tumor responses in different types of cancer. A variety of immune checkpoint inhibitor (ICI) proteins have been proposed as therapeutic targets in cancer treatments. The most clinically developed ones include programmed cell death protein 1 (PD-1), programmed cell death ligand 1 (PD-L1), and cytotoxic T-lymphocyte-associated antigen 4 (CTLA-4) [73]. In breast cancer, the TN subtype presents characteristics that make it more susceptible to immunotherapy, such as the increased infiltration of T lymphocytes and elevated levels of protein expression, such as PD-L1, in tumor cells and the immune system. This makes immune checkpoints promising therapeutic targets for the treatment of triple-negative breast cancer [74].

In general, schedules combining PD-1/L1 inhibitors and chemotherapy have shown greater success than ICIs as a monotherapy. For instance, the IMpassion130 trial, which included previously untreated mTNBC patients, demonstrated that adding atezolizumab to nab-paclitaxel resulted in a clinically significant improvement in OS by 7 months in the PD-L1 positive subgroup [75,76]. These results led to the FDA’s and European Commission’s approval of atezolizumab and nab-paclitaxel for mTNBC patients with PD-L1 positivity, marking the first immunotherapy approval in breast cancer. Subsequently, the KEYNOTE program showed that the use of pembrolizumab, a monoclonal antibody targeting PD-1, in combination with chemotherapy, led to clinically significant improvements in different clinical settings (KEYNOTE-355 and KEYNOTE-522 [77,78]).

Although immunotherapy has focused on the role of T cells in the adaptive immune system, combined immunotherapy strategies are currently being explored to overcome the apparent low immunogenicity of breast cancer. As this field continues to expand, there is a growing number of clinical trials that hold promise for the future. One area of growing interest in breast cancer is CAR-T therapy. Chimeric antigen receptor T cell (CAR-T) therapy is an immunotherapy approach that involves the transfer of T cells from the patient’s own peripheral blood. These T cells are isolated and modified outside the body (ex vivo) to express synthetic receptors capable of recognizing tumor-associated antigens. After this modification, these CAR-T cells are re-infused into the patient’s body as a cancer treatment [79,80]. Within breast cancer cells, altered expressions of several molecules are considered potential targets for CAR-T cell therapy. In recent years, several phase 1/2 clinical trials with different antibodies have been conducted. However, several challenges still persist, such as insufficient infiltration, the immunosuppressive environment, the lack of tumor-specific or tumor-associated antigens, and CAR-T cell toxicities [81].

#### 3.1.2. Modulation of TAMs for Therapeutic Applications

With a better understanding of cancer immunology, different strategies are being explored to modulate TAMs for therapeutic purposes. For instance, bisphosphonates, compounds with a high affinity for hydroxyapatite, are used in the treatment of bone diseases such as osteoporosis and Paget’s disease and for managing bone metastases. However, preclinical studies in murine models of breast cancer have suggested that they may also have an extra-skeletal therapeutic effect [29,82]. In this scenario, zoledronic acid, a bisphosphonate, attaches to microcalcifications present in breast tumors. Subsequently, it is phagocytosed via TAMs, inducing apoptosis and promoting the repolarization of M2 macrophages to M1. Although TAMs’ elimination improves survival in some preclinical cancer models, to our knowledge, complete tumor regression has not been achieved using bisphosphonates alone. Additionally, a study has attributed an additional role to TAMs in the processing of antibody–drug conjugates (ADCs) [83]. This study provided the first evidence of the potential contribution of TAMs to the efficacy of ADCs and introduced a less explored therapeutic avenue of ADCs specifically tailored to target TAMs.

### 3.2. Harnessing the Tumor Microenvironment Crosstalk as a Therapeutic Target

In ER+ breast cancer, estrogen provides a crucial mitogenic signal. Estradiol (E2) is synthesized in the ovaries and extragonadal sites, with extragonadal synthesis becoming dominant after menopause. In postmenopausal women, locally produced estrogen within fibroblasts and adipocytes represents the main source of E2. The enzyme aromatase plays a key role in E2 biosynthesis, making it an excellent target for microenvironment-directed therapy [84,85]. Third-generation aromatase inhibitors—anastrozole, letrozole, and exemestane—can inhibit aromatization in the whole body by up to 98%. Aromatase inhibitors have been shown to be even more effective than tamoxifen in preventing breast cancer recurrence in postmenopausal women [86]. These drugs are approved based on data from randomized clinical trials in both adjuvant [87,88,89,90] and metastatic settings [91,92,93], as well as in combination with other drugs like CDK4/6 inhibitors [94,95,96], olaparib [97], or everolimus [98], with similar efficacy and toxicity. Other hormonal treatments, which are not targeted at the microenvironment, act directly on the ER: selective ER modulators (tamoxifen and raloxifene) or selective ER downregulators (new SERDs).

In addition, in response to the hypoxic microenvironment, tumor cells release pro-angiogenic factors, such as vascular endothelial growth factor (VEGF) and fibroblast growth factor (FGF), which promote the sprouting of new blood vessels from nearby vessels. Understanding the mechanisms of angiogenesis has led to the development of anti-angiogenic therapies, a promising approach to cancer treatment. Bevacizumab, a monoclonal antibody targeting VEGF, has been approved for use in several cancers, including colorectal and lung cancer [99,100]. Since many neoplasms are highly vascular and rely on a solid blood supply to maintain cellular viability, bevacizumab has been studied in combination with various cytotoxic chemotherapy regimens. In 2008, its use was approved in combination with paclitaxel for HER2-negative mBC [101]. However, due to considerable controversy, this approval was revoked in 2010 due to safety concerns, including an increase in thrombosis cases, and a lack of overall survival improvement in a large number of patients [102]. Furthermore, a meta-analysis of 16 randomized controlled clinical trials showed that this agent, when used in combination with chemotherapy, entailed a higher risk of fatal adverse events related to treatment compared to chemotherapy alone [103]. Despite an extensive evaluation of the agent in different patient populations, a specific target population has not yet been identified. At present, the observed benefit in progression-free survival does not justify the additional toxicity, cost, and resource utilization associated with the use of bevacizumab, and its combination with a cytotoxic agent is only considered for selected patients [104].
ijms-24-16771-t002_Table 2Table 2Treatments and phase III clinical trials using the TME as a target in BC.Target DrugSettingReferences
Immune system

Inhibitor checkpoint

Atezolizumab (α-PDL-1)

mTNBC (first line PD-L1 >1%)
[78]
HER2+ (adjuvant high risk recurrence)
NCT04873362
mHER2+ (PDL-1+)
NCT04740918
Pembrolizumab(α-PD-1)

TNBC (NAC and adjunvant high risk recurrence) and
[80]
mTNBC (first line; CPS >10)
[81], NCT05382286
TNBC (with residual disease after NAC)
NCT05633654
ER+ (inoperable or mBC)
NCT04895358
Estrogen receptor signaling pathway

Aromatase inhibitor

AnastrozoleLetrozoleExemestane

Adjuvant BC ER+ (menopausal)
Alone [87,88,89]In combination with:iCDK4/6 [96]Olaparib [97]
mBC ER+ (any treatment line)
Alone [90,91,92]In combination with:iCDK4/6 [94,95,96] Everolimus [98]
Vascular system

Bevacizumab

mBC (first line)
[101]
BP102

mTNBC (basal-like immune-suppressed subtype)
NCT05806060
Adipose tissue

Breast Cancer Weight Loss Study (BWEL study)

Adjuvant BC (overweight and obese women)
NCT02750826
Impact of obesity

Adjuvant BC ER+ (aromatase inhibitors in postmenopausal women)
NCT01758146


As mentioned above, the interplay between CAAs and BC promotes tumor progression and contributes to treatment resistance. This suggests that targeting CAAs themselves or the secretome in CAAs may be a new strategy to further improve BC treatment effects. For instance, the PPARγ antagonist GW9662 inhibited the progression and metastasis of BC by selectively reducing PD-L1 expression in mouse adipose tissue and enhanced the antitumor effect of CD8+ T cells in homozygous BC models. Interestingly, a clinical trial is evaluating the impact of obesity on the efficacy of adjuvant endocrine therapy with aromatase inhibitors, and the BWEL study assesses whether weight loss in overweight and obese women may prevent BC’s recurrence.

## 4. Conclusions

Overall, the TME holds significant prognostic and predictive value in breast cancer. Its composition and interaction with tumor cells play a key role in tumorigenesis, progression, and treatment resistance in breast cancer. As stated above, tumors with a high content of immune cells often exhibit a better response to immunotherapies. Moreover, specific TME markers, such as blood vessel density or TIL infiltration, have been shown to be related to patient survival. Therefore, TME analysis can provide valuable information for selecting more effective treatments and identifying patients with a higher risk of recurrence. Additionally, understanding the TME can create opportunities for the development of personalized and combined therapies that target both tumor cells and their microenvironment. Ultimately, studying the TME is essential for improving the precision and efficacy of breast cancer treatments.

## Figures and Tables

**Figure 1 ijms-24-16771-f001:**
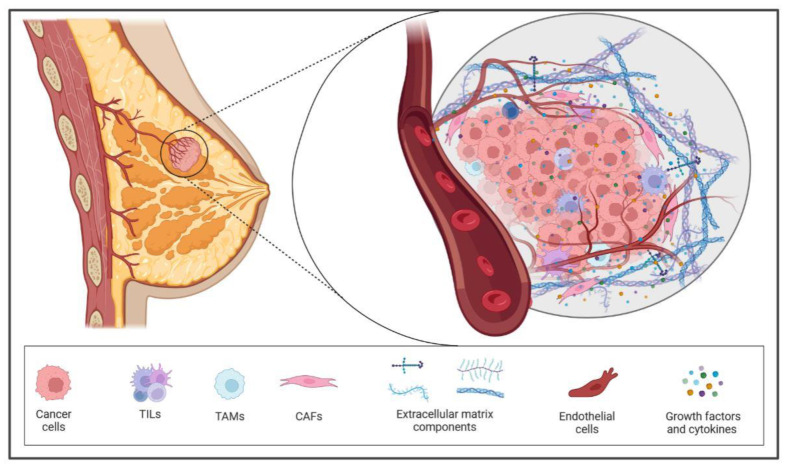
Breast tumor microenvironment. This figure offers a comprehensive view of the complex processes at play during breast cancer progression. The breast cancer TME represents a dynamic landscape crucial to understanding cancer progression. At the start of tumor formation, immune responses from cytotoxic T lymphocytes, M1 macrophages, and fibroblasts try to suppress tumor growth. However, breast cancer cells adapt and educate neighboring stroma cells to support their growth. Key players, such as cancer-associated fibroblasts (CAFs), are activated via cytokines, leading to the secretion of factors that support tumor development. Tumor-associated macrophages (M2 macrophages) also play a role by releasing pro-tumorigenic factors. As the tumor expands, it recruits regulatory T cells and other immune cells that hinder the immune response. All these orchestrated events within the TME contribute to breast cancer cells’ gaining mobility, invasiveness, and the ability to spread to secondary sites.

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
