# Peer review of "Targeting the Tumor Microenvironment in Breast Cancer: Prognostic and Predictive Significance and Therapeutic Opportunities"

_ijms, 2023, doi:10.3390/ijms242316771_

Round 1
Reviewer 1 Report
Comments and Suggestions for Authors
TME is a complex and heterogeneous network, composed of tumor, immune and inflammatory cells, cancer-associated fibroblasts, adipocytes, which secrete different factors, such as cytokines, chemokines, growth factors.
Breast tumor microenvironment is "mainly" composed of adipocytes, which are critical in linking obesity and breast cancer risk.
It is well known that adipose tissue is considered an active organ involved in the regulation of inflammation and metabolism through the production of many adipokines.Moreover, a large body of evidence suggests that the infiltration of proinflammatory macrophages into the adipose tissue, is responsable of the secretion of inflammatory mediators, sustaining a chronic low-grade inflammation, essential to produce a suitable TME for breast cancer development and progression.
Thus, I was surprised that the Authors didn't mention the adipocytes and their actions in this review.
An abundance of articles have been published on this issue!
Therefore, the Authors should include information about the role of adipocytes in breast tumour microenvironment and their possible contribution for the identification of therapeutic targets.
Author Response
|
Response to Reviewer 1 Comments
|
|
||||
|
Summary |
|
|
|
||
|
We appreciate your insightful comments and thorough review of our manuscript. Please find the detailed responses below and the corresponding revisions highlighted in track changes in the re-submitted files.
|
|
||||
|
Point-by-point response to Comments and Suggestions for Authors |
|
|
|||
|
Comments 1: Breast tumor microenvironment is "mainly" composed of adipocytes, which are critical in linking obesity and breast cancer risk. It is well known that adipose tissue is considered an active organ involved in the regulation of inflammation and metabolism through the production of many adipokines. Thus, I was surprised that the Authors didn't mention the adipocytes and their actions in this review. An abundance of articles have been published on this issue! |
|
||||
|
Response 1: We acknowledge the importance of the TME in breast cancer, which is indeed a complex and heterogeneous network involving various cell types, including adipocytes. We have revised the manuscript to address your concerns and now provide a comprehensive overview of the role of adipocytes in the breast TME. We have included information on the prognostic and predictive value of adipocytes and their potential therapeutic significance. In the reviewed version adipocytes can be found as a main component of the TME (see section 2.5, page 6) and its therapeutic significance is mentioned in section 3.2, page 10. Accordingly, we have updated table 1 and 2. We hope these revisions address your concerns and enhance the overall quality of our review.
|
|
||||
|
Response to Comments on the Quality of English Language |
|
||||
|
Point 1: English language fine. No issues detected |
|
||||
|
Response 1: We appreciate reviewer comments on the quality of English language. |
|
||||
Reviewer 2 Report
Comments and Suggestions for Authors
In this paper entitled « Targeting the Tumour Microenvironment in Breast Cancer : Prognostic and Predictive Significance and Therapeutic Opportunities » Domínguez-Cejudo María A et al reviewed extensively the research conducted for a better understanding of the role of tumor microenvironnement (TME) in breast carcinoma development, progression and metastasis. While genetic and epigenetic factors of tumor cells are mostly reported in the literature, this paper provides the readers with a nice up-to date of the least-well known role of TME in breast cancer (BC).
First the components of BC TME are reported (Fig 1, table1) and the functional and diversity of immune responses through the tumor-infiltrating lymphocytes and tumor-associated macrphages are presented, focusing on prognosis, outcomes prediction and therapeutic applications. Next the role of others stromal components (cancer-associated fibroblast, tumor vasculature) and a compilation of treatments and clinical trials targeting the TME are reviewed (Table 2).
This is a concise review, well-shaped and pleasant to read that should be very helpful for the non-experts to progress in this field. The conclusion highlights the interest of combined therapies targeting both the cancer cells and TME.
Is there some reports on microRNAs and TME that might be mentionned ?
It seems that original works from the team in this area are not cited.
Page 13, Ref 77, please write it with the authors’ name.
Author Response
|
Summary |
|
|
|
|
We sincerely appreciate your positive feedback on our manuscript. We are pleased to hear that you found the review concise, well-structured, and beneficial for non-experts in the field. Please find the detailed responses below and the corresponding corrections in track changes in the re-submitted files.
|
|||
|
Point-by-point response to Comments and Suggestions for Authors |
|
|
|
|
Comments 1: Is there some reports on microRNAs and TME that might be mentionned? |
|||
|
Response 1: we recognize the significance of this topic. While miRNAs play a crucial role in the regulation of the TME in breast cancer, we believe that a dedicated and detailed exploration of this subject would provide a more thorough understanding for readers. Therefore, we have decided to focus on the broader aspects of the TME in the current review.
|
|||
|
Comments 2: It seems that original works from the team in this area are not cited. |
|||
|
Response 2: Part of our team, including AGT, MC and JSB, are actively involved in clinical trials evaluating the addition of immunotherapy to breast cancer armamentarium. We have carefully reviewed the references and clinical trials cited and made the necessary additions to table 2 in the updated manuscript.
Comments 2: Page 13, Ref 77, please write it with the authors’ name. Response 2: Reference 77, updated to reference 82, now reads as follows: A. Mantovani, S. Sozzani, M. Locati, P. Allavena, and A. Sica, “Macrophage polarization: Tumor-associated macrophages as a paradigm for polarized M2 mononuclear phagocytes,” Trends in Immunology, vol. 23, no. 11. Trends Immunol, pp. 549–555, Nov. 01, 2002, doi: 10.1016/S1471-4906(02)02302-5.
|
|||
|
4. Response to Comments on the Quality of English Language |
|||
|
Point 1: I am not qualified to assess the quality of English in this paper |
|||
|
Response 1: Thank you for your comment regarding the quality of English language |
|||
Round 2
Reviewer 1 Report
Comments and Suggestions for Authors
I thank the Authors for implementing the manuscript with the points raised.